# Microfluidic Organ-on-A-chip: A Guide to Biomaterial Choice and Fabrication

**DOI:** 10.3390/ijms24043232

**Published:** 2023-02-06

**Authors:** Uyen M. N. Cao, Yuli Zhang, Julie Chen, Darren Sayson, Sangeeth Pillai, Simon D. Tran

**Affiliations:** McGill Craniofacial Tissue Engineering and Stem Cells Laboratory, Faculty of Dental Medicine and Oral Health Sciences, McGill University, 3640 University Street, Montreal, QC H3A 0C7, Canada

**Keywords:** microfluidics, Organ-on-A-chip, 3D printing, biomaterials, microfluidic fabrication

## Abstract

Organ-on-A-chip (OoAC) devices are miniaturized, functional, in vitro constructs that aim to recapitulate the in vivo physiology of an organ using different cell types and extracellular matrix, while maintaining the chemical and mechanical properties of the surrounding microenvironments. From an end-point perspective, the success of a microfluidic OoAC relies mainly on the type of biomaterial and the fabrication strategy employed. Certain biomaterials, such as PDMS (polydimethylsiloxane), are preferred over others due to their ease of fabrication and proven success in modelling complex organ systems. However, the inherent nature of human microtissues to respond differently to surrounding stimulations has led to the combination of biomaterials ranging from simple PDMS chips to 3D-printed polymers coated with natural and synthetic materials, including hydrogels. In addition, recent advances in 3D printing and bioprinting techniques have led to the powerful combination of utilizing these materials to develop microfluidic OoAC devices. In this narrative review, we evaluate the different materials used to fabricate microfluidic OoAC devices while outlining their pros and cons in different organ systems. A note on combining the advances made in additive manufacturing (AM) techniques for the microfabrication of these complex systems is also discussed.

## 1. Introduction

Microfluidics refers to the science of fluids at the micron scale. Microfluidic devices manipulate fluids through the microchannels in the platform. Today, microfluidics is still seen in inkjet printers, one of the first technologies to use them. The first microfluidic devices were produced using silicon and glass, as they were born in the field of microelectronics. However, they are now commonly made using PDMS (polydimethylsiloxane) due to their relatively low cost, ease of use, and biocompatibility [1]. Several manufacturing processes are used in the creation of microfluidic devices. Soft lithography remains a standard production method, but 3D printing is becoming more commonly employed as technology improves [2]. At the micro-scale, there are several benefits of the unique properties that fluids take on in microenvironments. The laminar flow of the fluids, concentration gradients, and the high surface area to volume ratio of the channels replicate the fluid flow seen within biological systems [3]. For this reason, microfluidic devices create an appealing platform to study and replicate human organ systems closely.

Organ-on-A-chip (OoAC) devices employ microfluidics to accurately replicate an in vivo environment. The microfluidic channels provide a basis for creating a physiologically similar environment to culture the organ cells of interest [4]. Another method for modelling and studying organs are organoids, a 3D cell culture system where the cells self-organize and differentiate into organ-specific tissue [5]. While a handy tool, organoids lack vital aspects in the human microenvironment. In vivo characteristics such as vasculature, oxygen gradients, chemical gradients, and mechanical stress can be recreated by combining these 3D cell culture techniques with the microfluidic platform. Furthermore, in some cases, the mechanical strain in microfluidic channels is crucial in developing physiologically relevant cell culture models and their resulting molecular characteristics [6]. These devices capture the essential functions and microarchitecture of the organ to be investigated [7]. OoAC provides a promising method for drug testing and discovery as the disease states can be replicated in addition to normal organ function with versatility that animal models or 2D cell cultures cannot capture as accurately [8]. Numerous organ models have been developed, with major organ systems such as the lung, gut, heart, and kidney being just a few OoAC devices currently used in drug discovery [9,10,11]. This combination of 3D cell culture with microfluidic technology confers several advantages as it allows for similar cell–cell/tissue–tissue interactions and biochemical and physical environments present in vivo.

For medical applications, OoAC devices have a wide variety of applications. Broadly, OoAC devices are classified by their respectively modelled organ; however, these devices can be further subdivided into modelled diseases. For example, lung chip devices have replicated disease states such as cancer, pulmonary edema, and cystic fibrosis [12,13,14]. Another example is the fabrication of bone marrow-on-a-chip (BMoC) by Tserepi et al. (2020). The authors demonstrated a scaffold-free BMoC device as a study platform for systemic lupus erythematosus (SLE), which is illustrated in Figure 1. The study marked a preliminary step for a complete development of perivascular system-on-chip [15]. OoAC platforms also provide the opportunity for personalized medicine as the human-induced pluripotent stem cells allow for the investigation of therapies that are tailored to specific patients or populations [16,17]. The global OoAC market has been steadily growing in recent years, primarily due to its drug discovery and testing applications, with an expected compound annual growth rate of 30% over the next five years [18]. Furthermore, it is predicted that the development of treatments for COVID-19 and research into the effects of the disease will lead to the growth of the OoAC market, which has indeed been the case as researchers gain an understanding of SARS-CoV-2 and how it affects the human body. A recent study has repurposed the vasculature-on-a-chip model by culturing the endothelial cells in the presence of SARS-CoV-2 to further understand the pathophysiology of the viral infection and suggest potential therapies [19]. Another study repurposed the airway-on-a-chip platform similarly to gain insights into how the virus affects multiple systems, elucidating the interplay between vasculature and respiratory organs in a single chip [20]. To further understand the systemic effects of the virus, researchers have manipulated the OoAC device to capture different segments of the lung, incorporating the nasal passage, the mid-bronchial airway, and the tissue of the alveolus [21]. An advantage of OoAC is the ability to easily monitor the microenvironment in real time via integrating sensors into the platform [22]. As the virus evolves and SARS-CoV-2 variants emerge, microfluidic platforms can also be used for the screening of new variants and testing potential treatments [23]. Investigation into this viral outbreak has led to the further development of existing OoAC models. For example, the alveolus-on-a-chip platform has been improved, incorporating a central collagen channel to act as an extracellular matrix, adding an increased level of complexity [24]. These studies capture research on how these biomimetic platforms are integrated into investigating today’s most pressing medical issues.

This review presents the various materials and evaluates the properties that make them ideal for the fabrication of OoAC devices. Furthermore, recent advances in elastomeric synthetic materials, thermoplastic synthetic materials, and natural biomaterials are compiled and compared. Lastly, the fabrication strategies employed in producing OoAC devices are assessed to provide a comprehensive picture of the current state of OoAC devices.

## 2. Biomaterial Selection Criteria

One of the first critical steps in OoAC device design is the selection of the biomaterial that will form the basis of function and tissue fabrication in the chip system. Current biomaterials commonly used in many OoAC systems satisfy many specific criteria (Table 1), further outlined in this paper. Biomaterials can be classified into two general categories: natural polymers and synthetic polymers. Natural polymers are materials derived from natural sources, such as extracellular matrix (ECM) components, plants, insects, or other mammals [25]. The characteristics of these materials typically match those of their source and thus have high biocompatibility, making them very useful in many OoAC applications (i.e., tissue engineering, bio-sensory development, and drug-delivery systems) [25]. So far, several natural biomaterials have been developed (Table 1) and are used routinely in OoAC devices fabrication. These materials are largely sought due to their superior properties that support cell proliferation and adhesion, allow mechanical stimulation, and fulfil rheological dynamics and other complex modifications. However, natural polymers suffer from batch-to-batch variation, with significant differences depending on the type and quality of the natural source, and also require a careful sterilization and purification process [25]. To overcome the shortcomings of natural polymers, synthetic polymers have been developed to finely control their physical and chemical properties and produce materials that can go beyond the bounds of naturally derived materials with a minimal batch-to-batch variation. Synthetic materials often lack cell-adhesion ligands on their surfaces and thus require chemical modification to promote tissue development [25]. Within these two categories, there are several variations and aspects of each material that must be assessed on a case-by-case basis. The importance of each of these criteria is defined based on the properties of the tissue the OoAC is attempting to mimic. Table 1 outlines the properties of commonly used natural biomaterials that have already been well-characterized and used successfully in viable chip systems. Moreover, Figure 2 summarizes some of the popular OoAC systems, along with partnered biomaterials and their properties.

### 2.1. Biocompatibility

A biomaterial must be biocompatible to create an environment where tissues and their constituent cells can grow. A biocompatible material is non-toxic to cells, so there is no damage or stress to the OoAC system that would influence results or the accuracy of the model [25,57]. A biocompatible material also implies that the pH, oxygen, and carbon dioxide levels are compatible with the chosen cell type for tissue development. Additionally, the preferred material must not trigger toxic reactions with the medium used for the cell culture [69]. In most cases, the biomaterial should be able to support a continuous supply of oxygen to the cells, have a balanced pH system of approximately 7.4 [70,71], maintain a 4–10% carbon dioxide level, and promote cell adhesion for a functional environment [57,70]. The importance of biocompatibility is apparent in all OoAC systems, especially if the end goal is to achieve a thoroughly combined system with the natural body.

### 2.2. Biodegradability

Biodegradable materials can also provide several benefits but may lead to certain limitations to the OoAC system. If the system’s goal is to be eventually integrated into the natural extracellular matrix, then the biomaterial must also be biodegradable. The rate of degradation must also be inversely proportional to the production of biological tissues by the cells to properly balance and maintain homeostasis within the system [72]. The by-products of biodegradation must be considered as well, as they must be easily removed from the system to prevent any adverse effects. For example, the standard synthetic materials such as polylactic acid (PLA) or poly(glycolic acid) (PGA), and their copolymers have acidic by-products that could affect the system’s natural pH, thus affecting the natural behaviour of the cells in an undesirable way [73,74]. The biodegradable properties of the biomaterial can also be used to an advantage for small molecule testing. By assessing the kinetics and degradation rates of the material, a more accurate model can be developed for drug testing and assessing drug response in specific organ models [75]. Characterizing a material’s biodegradability, and its potential functions and limitations, can be used to advance the physiological relevance of an OoAC and thereby prevent any unwanted effects caused by toxic by-products [57].

### 2.3. Mechanical Properties

Natural and synthetic biomaterials can vary significantly in their mechanical properties, which provides a wide range of options for various organ tissue functions. The biomaterial selected must have adequate mechanical properties to match the mechanical characteristics of the target organ. In the context of microfluidic chips, a few critical mechanical properties may have a significant effect on the outcome:The biomaterial’s response to fluid shear force influences how the cells develop, grow, and survive within the chip. Specifically, it affects the alignment of mini-organs and cells by altering the polarity axis, and the physical pressure also activates signalling molecules within and on the cell surface [76].Microfluidic chips primarily undergo laminar flow, which controls their mixing rate but have slow rates of diffusion [32]. This creates a stable gradient within the OoAC chip that can influence biological processes, such as cell migration and movement, or even simulate developmental processes such as angiogenesis [76].The biomaterial’s dynamic behaviour in response to mechanical stress can also be beneficial to mimic tissues or organs that undergo constant loading and physiological changes (i.e., bone, muscle, cartilage, blood vessels) [76]. These characteristics are also typically defined by mechanical properties such as Young’s modulus, Poisson’s ratio, and ductility. They can be used to select a biomaterial which mimics the desired tissue behaviour more accurately [25,77].

### 2.4. Sterilization Techniques

The biomaterial must be appropriately sterilized to prevent damage or cross-contamination of the OoAC chip. It is essential for natural polymers since they are typically derived from animals or plants, which can contain harmful chemicals or microbes that could influence the results [2]. However, the properties of each biomaterial dictate which sterilization technique can be used, which may sometimes be the limiting factor in the selection of a biomaterial. Damage during the sterilization process can result in leaks in microchannel walls, which influences the chip’s behaviour [2] or may affect the mechanical properties of the biomaterial [67]. For example, PMMA and PC cannot undergo conventional autoclaving techniques, as they require exposure to high temperatures, which would melt the two biomaterials [2]. Overexposure to certain chemicals can also result in chemical absorbance in the chip walls, which may leak out during testing and influence cell behaviour within the chip. PDMS soaking in ethanol for an extended period will result in the material dissolving or even absorption of the chemical [57]. When selecting a biomaterial, assessing the available resources and the feasibility of using the material and its properties is essential.

### 2.5. Surface Treatment for Cell Patterning

To combat the challenge of mimicking the 3D environment that promotes cell growth, surface treatment techniques on biomaterials have been developed to enhance cell attachment and proliferation [76]. Surface treatment or cell patterning techniques alter the surface geometry of the chip’s material to control the arrangement of cells that influence their differentiation and growth [76] and can even enhance cell adhesion [2]. In some cases, physical or chemical signals may also be needed with alignment factors to stimulate further cell growth, such as highly organized myocardial tissue with electrical stimulation [76]. In the process of surface patterning, key parameters include feature resolution size, throughput surface area range, background contrast, bioactivity, and shelf-life [78]. Considering these parameters, a specific surface technique can be selected for a particular cell type to optimize growth and control tissue functions [79]. Some examples of surface patterning techniques include:Pluronic acid treatment to prevent 3D spheroid or organoid culture attachment in certain areas of the chip and to control the 3D tissue architecture and thus its function [80].Protein and ECM coating (i.e., Matrigel) to promote monolayer intestinal cell attachment and model the epithelial layer of the gut [81].

Each of these techniques can vary depending on the intended application and goal of the OoAC system. Similar to the limitations of sterilization techniques on biomaterials, the available surface patterning techniques must also be assessed before selecting a biomaterial for OoAC designs. 

## 3. Recent Advances in Biomaterials for OoAC Systems

Though the OoAC field has only been around for ten years of history, there is a rapid rise in the number and size of companies commercializing these OoAC devices (∼30 companies in 7 years), indicating a considerable market lying ahead in OoAC research [82,83]. Fuelled by the potential of transforming into a drug discovery platform, researchers are optimizing the OoAC to mimic the structure and extracellular environments of different tissues/organs meticulously [16,76,84].

PDMS was the material of choice in most of the pioneering work in microfluidic device development and is still one of the most widely used materials for OoAC fabrication [82]. It offers excellent properties, including biocompatibility, optical transparency, low autofluorescence, high elasticity, excellent oxygen permeability, and very good deformability [25,57,82]. However, researchers have been searching for alternative fabrication materials or coatings in recent years due to several drawbacks of PDMS, including the absorption of tested drugs and small hydrophobic molecules from culture media [85]. This leads to a significant possibility of misinterpretation of drug toxicity and efficacy when various drugs are tested on the same OoAC platform.

### 3.1. Elastomeric Synthetic OoAC Materials

The search for more affordable alternative OoAC materials with ease of fabrication has been ongoing for many years. Studies show that polyester-toner microfabrication can be an excellent alternative chip material due to its low-absorption, soft-elastic, and biocompatible characteristics [86]. Urbaczek et al. generated a vein-on-a-chip made of polyester and toner. The additional application of oxygen plasma and fibronectin increasingly supported the adhesion and proliferation of endothelial cells [87]. It also demonstrated that polyester materials could provide good mechanical strength for soft tissue regeneration with low toxicity and optical transparency. Recently, poly- (lactic-co-glycolic acid) (PLGA), a biodegradable and biocompatible material previously used for drug delivery, has been investigated for its potential as microchip material. Gao et al. fabricated a lung-on-a-chip with PLGA electrospinning nanofiber as the chip substrate with controllable thickness [88]. Lung cancer cells and fibroblasts were 3D co-cultured on the porous PLGA membrane with PDMS as the cover to simulate the human alveolar microenvironment [88]. Poly (octamethylene maleate (anhydride) citrate) (POMaC) is another soft biodegradable elastomeric platform biomaterial applied in microchip fabrication, whose mechanical properties are close to a wide range of soft biological tissues [89].

### 3.2. Thermoplastic Synthetic OoAC Materials

Poly (methyl methacrylate) (PMMA) is another great candidate to replace PDMS due to its features of recyclability, biocompatibility, excellent mechanical properties, and low permeability to small molecules [90,91,92]. Bhise et al. designed a bioreactor consisting of multilayers of PDMS and PMMA and included three culture chambers connected by fluidic channels [84]. Hepatocyte spheroid-laden hydrogel could be bio-printed directly within the chambers. These chambers and channels were fabricated by casting PDMS around a laser-cut PMMA mould [84]. Polyetheretherketone (PEEK) has been recently considered a potential substitute because its fabrication is inexpensive and high-quality [93]. However, its low degradation rate and strength limits its application in chip fabrication [94,95]. To overcome these disadvantages, researchers have tried to blend PEEK with other biomaterials, such as polyglycolic acid (PGA), which has excellent degradability and biocompatibility [96]. It has been proved that modifying the percentage of PGA added to PEEK scaffolds could alter their degradation rates. PEEK/PGA blend scaffolds have thus shown promise for OoAC applications.

### 3.3. Natural OoAC Materials

To recapitulate complex tissues accurately, the biomaterials chosen should be able to better support the sustainability and functionality of cells or spheroids in the long term. PDMS, along with the OoAC platform, has many limitations, especially when rebuilding complex organs such as the lung, heart, and brain. Natural biomaterials are crucial to consider as part of the organ-on-chip scaffold due to their superior biocompatibility, bio-adhesion, biodegradability, and non-toxic nature [97,98]. Simulating tiny alveoli with similar dimensions, Zamprogno et al. created a lung-on-a-chip with a biological, stretchable, and biodegradable membrane using collagen and elastin [99]. Compared to PDMS, this naturally derived membrane had several merits: it did not absorb small molecules such as rhodamine-B, they are biodegradable, and the degradation rate, thickness, composition, and stiffness could be easily tuned [99].

A microfabricated bioreactor was designed to mimic natural cardiac bundles in vitro using cardiac bio-wires [100]. Type I collagen was selected as the main gel matrix, which is one of the primary components of native myocardium. The collagen-based cardiac bio-wires remained stable in the bioreactor for weeks, with the mechanical support provided by the suspended templates made of either silk suture or polytetrafluoroethylene (PTFE) micro-tubing [100]. Nashimoto and colleagues presented a microfluidic device made of PDMS and fibrin–collagen gel culturing 3D cellular spheroids with a vascular network [101].

Gelatin-methacryloyl (GelMA) is a semi-synthetic hydrogel which consists of gelatin derivatized with methacrylamide and methacrylate groups [102]. The gelatin part of GelMA hydrogels provides cells with an ideal biologically adhesive environment. Adding methacryloyl contributes to quick photo-crosslinking, shape fidelity, and stability at physiological temperature, while gelatine’s biocompatibility and degradation properties remain unaffected [103,104]. Bhise and colleagues have recently designed a liver-on-a-chip made of GelMA hydrogel encapsulating 3D hepatic spheroids, where the engineered hepatic construct remained functional during 30 days of culture [84]. Yang et al. reported that multicellular vascular channels were regenerated in their 3D-printed design with a GelMA hydrogel scaffold [43].

Without a doubt, such intricate processes that rely on many factors can be challenging to control tissue development. However, state-of-the-art research shows promising advances in overcoming this challenge by designing OoAC with organ-specified structure using synthetic or natural materials or a combination of both. In the future, we are looking for further development and standardization of using scaffolds for specific cell types, making it easier to compare the resultant data from different research groups [57].

## 4. Fabrication Techniques for Microfluidic Systems

Although microfluidic OoAC is still in its infancy, scientists have developed quite a diverse repertoire of microfabrication techniques to manufacture an OoAC, which is shown in Table 2.

### 4.1. Lithography

The lithography technique, widely used in the printing industry nowadays, was launched in 1796 [163]. Since then, the lithographical fabrication of OoAC has been extensively investigated and developed. Depending upon the type of materials used, the fabrication techniques vary. Currently, the two primary methods are photolithography and soft lithography.

The first one, which is also named optical lithography or UV lithography, is a method that utilizes light, mostly UV light with a wavelength of 250–435 nm, to transfer three-dimensional designs from a template onto a thin photo-sensitive (photoresist) film attached over a silicon substrate [112,164]. When exposed to light, the photo resistant parts change their chemical structure and become solidified at the end of the process. Later, the unexposed areas will be baked and washed away in the chemical bath development, and the hardened parts will remain intact to create a negative or a positive mould. After that, the micropattern will be introduced to an etching agent and engraved into the solid wafer [164]. This technique’s capacity to produce microscopic structures (sub 250 nm) with high fabrication accuracy and relative efficiency gives it a strong advantage. Despite its numerous benefits, the method does possess some drawbacks. For example, the equipment for this technique is exorbitant, and it requires a cleanroom for the photolithographic machine to operate [111]. Recently, to eliminate those disadvantages, Kasi et al. (2022) used a simplified process called maskless photolithography [112]. The novel approach developed a more time- and cost-effective and cleanroom-free fabrication using 375 nm UV light for a negative photoresist (SU-8) mould. The authors reported that grayscale photolithography could produce microstructures with different widths and heights in a single-step operation. They concluded that the findings would promote the manufacture of variable OoAC platforms in the future [112].

Regarding soft lithography, this technique is an improvement of photolithography, as it is economical, easier to replicate, and can be applied to a broader range of materials [118,165]. The steps are similar: the wafer is coated with a photo-resistive layer and then cross-linked with UV light. After dissolving the unexposed part, a developer will be used to create a 3D micropattern on the wafer to build a negative mould. In the fabrication of some microfluidic devices, elastomers are often utilized. First, a cross-linking agent mixed with PDMS prepolymer will be degassed to remove air bubbles, poured into the fabricated mould, and then cured at high temperatures. Eventually, the PDMS device will be detached from the mould and micro-punched to create inlets and outlets for fluidic flow. Compared to photolithography, this technique undergoes fewer steps and can manufacture multiple OoAC devices in a shorter period [165]. Within the family of soft lithography are various patterning techniques such as replica moulding (REM), micro-transfer moulding, phase-shifting edge lithography, and nano-transfer printing [118,166,167]. REM, which has recently emerged as an improved method for fabricating microfluidic devices using PDMS, involves three basic steps: (1) constructing a desired master, (2) transferring the master’s design on the PDMS, and (3) transferring the PDMS’s pattern back into a duplicate of the original master by UV/heat curing liquid the pre-polymer or thermosensitive resin [118]. REM is typically applied for sophisticated structures on a nanoscale (less than 100 nm) and produces large quantities of replicas from one mould. Therefore, this technique has been utilized for multiple applications such as cell shaping, DNA sorting, pressure sensor, other healthcare devices, and rehabilitation monitors [168,169,170,171].

### 4.2. Injection Moulding

Injection moulding is a well-established technique that has been long used for mass polymeric fabrication. The principle of injection moulding is relatively straightforward and involves four stages: clamping, injection, cooling, and ejection [122]. First, two halves of the mould, which were previously created, are tightly clamped together. Then, the plastic material is heated into liquid and injected into a mould to form the desired shape. At this stage, the injection rate, such as pressure, power, and volume, can be modified and controlled to achieve optimal results. Next, the cooling process begins after the products inside the mould solidify. The material may shrink to some extent during this stage, which depends on the mould structure, moulding conditions, and polymer properties. Finally, the mould is opened and the product is ejected by an ejection unit. This process must be carried out with great attention and proper force, as excessive tension will cause damage to the final products [172]. Recently, researchers have embarked on a journey to use the technique in microscale yet high-volume production of OoAC devices [57]. Even though the principle of this technique may sound simple, many challenges need to be addressed when applied to the microfabrication of OoAC devices. Some obstacles include finding a suitable material to produce unique features, creating sophisticated nanoscale geometries, and developing an appropriate injection rate to ensure the products are of the highest quality [173]. To improve the limitations of this technique, Lee et al. (2018) developed a novel injection-moulded plastic array 3D scaffold (IMPACT), which incorporates the microfluidic structure into the design and embedded with human umbilical endothelial cells (HUVECs) and fibroblasts [172]. The authors demonstrated that injection moulding could generate cell-laden and ECM-mimicking scaffolds better than conventional PDMS-based techniques. While PDMS devices have some limitations with 3D design and fabrication, IMPACT offers the freedom of multi-level designs, mass producibility, and space efficiency. Moreover, it is also reported to possess an excellent capacity for drug delivery, angiogenesis, and vasculogenesis. Regarding the distinction between IMPACT over PDMS platforms, researchers believe that IMPACT will further be applied in numerous biomedical areas in the future [172].

### 4.3. Hot Embossing

Hot embossing lithography (HEL) is an efficient replication technique to fabricate microstructures, especially microchannels in microfluidic devices. This method uses a master mould with high pressure and temperature to shape a thermoplastic material into a desirable structure [174]. The manufacturing process usually includes four stages. First, the solid material is heated into a viscous liquid. Next, the master mould is pressed into the polymeric solution. Then, the mould is firmly placed until the hot solution cools down and hardens. In the final step, the newly fabricated structure is carefully removed from the mould. Compared to other techniques, hot embossing possesses multiple advantages, such as cost-effectiveness, high efficiency, and excellent structural precision [133]. The material shrinkage and damage using this technique is often less significant than the other methods, as the temperature change and frictional forces are much lower [175]. However, there are still some significant challenges that scientists must face when using this technique. Firstly, it is difficult to control the optimal process parameters such as material thickness, applied pressure, and heated temperature to secure and completely fill the cavity of the master mould with the polymeric material, to achieve the highest precision. Moreover, the force for demoulding also requires optimization; once the hardened material is removed from the mould, care should be taken to prevent distortion of the newly formed structure [131]. A study by Cogun et al. (2017) investigated the hot embossing technique on poly(methyl methacrylate) (PMMA) microfluidic channels [132]. The authors developed microchannels with varying widths and depths to observe differences in the replication rates and symmetrical manifestations. They discovered that the embossing temperature was the critical factor during manufacturing and the skewness of the device depending on the distance of the cavity from the centre of the PMMA structure. These results demonstrated that the flow property and the heated temperature significantly influenced the skewness of the substrate. The optimal parameter for this technique was reported at 115 °C, 10 kN, and 8 min for a microchannel of 56 um width and 120 um depth [132].

### 4.4. Etching

The etching technique is a subtractive process. The desired material will first be coated with an etch-resistant layer, which must be patterned according to the designed structures. The uncoated areas will then be etched with physical or chemical agents. For the chemical agents, the technique is called wet etch, and plasma-based dry etch uses the physical etching method. In the wet etching method, the material is removed from a wafer by liquid chemicals, which consists of three primary stages: the occurrence of etching reactions, the removal of diffused liquid etchant, and the diffusion of the by-products from the reaction [176]. Etchants can be hydroxides of alkali metals such as potassium hydroxide (KOH) and sodium hydroxide (NaOH), or acidic solutions such as nitric acid (HNO_3_) and hydrofluoric acid (HF) [142,177,178]. Etchants can engrave into the substrates isotropically or anisotropically. An isotropic etch will remove the material uniformly in all directions, while an anisotropic one will cause the structure to corrode vertically. Both types of etching can produce different geometrical channels for microfluidic devices, depending on the substrate materials and channel designs. Yet, it is still challenging to construct microchannels with high precision using the etching method [176]. This technique is an effective way of mass fabricating a microfluidic platform. In the dry etching method, the masked structures are exposed to plasma or ion flows, which often contain reactive etchant gases such as oxygen, nitrogen, and fluorocarbons. The process is conducted under a wide range of pressure (ranging from 2 to 7 mTorr), temperature (−140 °C–20 °C), and power (800–1500 W) [179]. A combination of dry and wet etch is called reactive ion etching (RIE), often utilized to achieve the highest resolution. RIE, an anisotropic process using reactive gases, such as CF_4_, SF_6_, and BCl_2_, has been one of the most popular etching techniques in research and industry, as it can transfer state-of-the-art pattern designs with vertical sidewalls into the substrates [180]. Despite some drawbacks, such as time consumption, exorbitant cost, low etch selectivity and low etching rate, RIE can fabricate many feature designs and geometries up to thousands of micro- and nanometres [181]. The impressive characteristic of RIE has caught much attention worldwide, which hopefully will be trending in the application of integrated sensors and microfabrication of microfluidic devices, especially OoAC platforms soon.

### 4.5. Laser Cutting Process

The laser and plotter cutting processes are two popular methods to create microfluidic channels, often for plastic, glass, paper, and pressure-sensitive adhesive tape materials [148]. In laser cutting, maskless fabrication methods such as CO_2_ laser, ultrafast laser, excimer laser, or UV laser can often obtain an exact cut of multiple layers. The current laser cutting systems are connected to galvanometer-based scanners and computer-aided design (CAD) systems, which can construct complicated micro-patterns on the substrate with various depths and widths. The development time for the whole process is relatively short, which takes around 30 min to complete. Yet, this laser causes the material to decompose through heat and leaves many pyrolysis by-products at the end of the process, which may block the channels in microfluidic devices. Although the laser cut possesses the advantages of non-contact carving and excellent resolution, the extortionate equipment and the need for high-pressure compressed air or water blow to wipe out the burn residue are some of the drawbacks of using laser cutting [150,181,182]. For the plotter cutting method or xunography, non-thermal tools such as knife printers are used to cut the microstructures. This can help avoid pyrolysis or any chemical debris, requiring no cleaning afterwards. The overall process of both laser cutting and plotter cutting techniques is relatively short and does not require a cleanroom environment for fabrication. Recently, they have been widely commercialized as a tool for fabricating microchips in the industry, owing to the rapid prototyping worldwide user database leading to significant technical advancement of both systems [150].

### 4.6. 3D Printing

In recent decades, 3D printing has emerged as an innovative field with multiple applications in machinery [183,184], businesses [162], and the biomedical industry [185,186]. This technique has also gained much attention in microfluidic domain, allowing scientists to engineer complex OoAC platforms that are difficult to design with traditional methods such as etching or moulding. This is an addictive manufacturing technique for the fabrication of 3D models, which means that the models are created by layers of materials added on top of one another [93]. The system primarily consists of a 3D printer, print materials, software, and on-demand manufacturing services. The printer can move in 3D space to construct 3D devices, which is supported in three axes XYZ. The X- and Y-axes are responsible for the 3D printer’s lateral movement, while the Z-axis represents the vertical movement of the nozzle. Every 3D printing procedure commences with a computer-aided design (CAD) model from software. Then, the materials are extruded from the nozzles. 3D printers use laser light to harden the photosensitive resin from a liquid state or melt polymer particles at high temperatures to create models. Depending on the type of printing system, the products might need post-processing curing to stabilize mechanical and physical features and to be cleansed with alcohol or compressed air to eliminate plastic residue on their surface [187,188,189]. Three of the most well-developed 3D printing types are stereolithography (SLA), selective laser sintering (SLS), and fused deposition modelling (FDM). Using UV laser beams for photosensitive thermoset polymeric curing, SLA has been one of the most popular 3D printing technologies. By modifying the beam’s direction, the polymer’s crosslinking process can be controlled and adjusted according to the manufacturers. SLA method is well-known for its capacity to fabricate products with state-of-the-art accuracy, delicate features, smooth surface resolution, and watertight tolerances [190,191,192]. Regarding SLS, this powder bed fusion technique uses a high-power laser to heat-treated polymeric particles to form the desired structures. Nylon, polystyrenes (PS), polyaryl-etherketones (PAEK), and polycaprolactone (PCL) are the familiar choices of material for this technique [159,193,194]. Similar to SLA, SLS is also ideal for complicated microstructures and geometries. Last but not least, the FDM technique, often called fused filament fabrication (FFF), was first developed in 1989 and has been widely commercialized worldwide ever since [158]. The process starts with a heated thermoplastic ink extruding from a nozzle to form a 3D structure. After releasing the liquid material into the desired area, the material will be solidified. The process keeps building the model layer-by-layer to the end [158]. This additive method is ideal for basic proof-concept or simple 3D models because of its simplicity, cost-effectiveness, and utilization of various materials. Typical heights of FDM layers range from 50 to 400 microns. The shorter the layer, the more accurately the structures will be created; however, achieving the goal will take more time and cost. In general, FDM cannot compete with SLA and SLS on precision, time, and cost-efficiency, and it is often not recommended for printing sophisticated microfluidic designs and structures [159].

## 5. Conclusions and Future Perspectives

Over the last few decades, conventional 2D and 3D cell culture systems have seen a significant improvement in terms of reconstituting organ-specific functions. This growth can be safely attributed to diverse cell sources, well defined culture conditions, and the development of novel ECM substitutes for sustaining cellular and tissue-specific functions in vitro. However, the static microenvironment surrounding these culture systems limits our ability to directly extrapolate these findings into their clinical counterparts without being interfered with by the long and time-consuming in vivo studies. Recently, OoAC microfluidic systems have emerged as an alternative, highly sought-after platform owing to their ability to bypass the confines of these static culture systems. These microfluidic devices aim to closely recapitulate an organ system by accurately defining physiological parameters such as cell-tissue interactions, mechanical cues, and fluid perfusion. In addition, they provide both immune and vascular components to perform organ-level functions.

Much like the success of the 3D culture systems, the OoAC microfluidic chips were influenced mainly by the development of natural and synthetic biomaterials complemented by the advent of versatile microfabrication techniques. The ease of fabricating PDMS chips with sub-micron scale channels to reconstruct microtissues or mini organs using conventional soft lithography has opened a large avenue in the field of biotechnology focused on developing organ chips. As a sophisticated cell culture model, OoAC devices allow us to perform real-time, dynamic studies on patient-derived cells and tissues. They are widely used in testing drug efficacy and toxicity, performing disease modelling while tracking disease progression, and assessing the pharmacokinetics of a metabolite all embedded into an integrated chip. The rapid growth of microfabrication techniques has allowed the development of several complex OoAC systems, such as the lung, heart, liver, and kidney, on a chip, to name a few.

Nonetheless, several challenges still prevail, limiting the transition of using on-chip platforms for accelerated drug discovery, especially in cases of rare diseases where in vivo models are challenging to develop, and clinical trials are bound by regulatory bottlenecks. In addition to the social challenges, OoAC also suffers from the demerits of often being low to medium throughput as it lacks the versatility of 2D and 3D culture (such as 2D/3D assays conducted in 96-well, 384-well plates). From a biomaterial perspective, ongoing concerns point toward the cons associated with using PDMS chips due to issues with their absorption profiles during drug/metabolite screening. Some hydrophobic drugs, such as sex steroids and nifedipine, are strongly absorbed by PDMS, affecting the results of the pharmacodynamic kinetics and drug efficacy studies. To overcome these challenges, alternate materials such as PMMA, polycarbonate, cyclic olefin polymers, polylactic, etc., were introduced. However, not all of them provide sufficient biocompatibility and thus are not sustainable for developing biomimetic cellular constructs.

Another major consideration in the development of microfluidic OoAC systems is their acceptance as a robust platform to replace animal models. This includes concerns of placing these human chips into the drug development and clinical trial pipelines. At present, rodent models and higher order mammals are the keystone players in successfully translating preclinical results into human trails. However, it is still challenging to develop highly specific disease models, such as rare diseases which have recently been improved with the recent advance in gene editing and transgenic models. Given this scenario, developing an acceptable microfluidic system, which not only recapitulates the disease, but is widely accepted globally by pharma companies and regulatory boards. is quite difficult. Most sophisticated and upcoming organ chips are developed by research teams in educational settings who focus on a particular type of disease. These models are usually unavailable across other research groups globally in order to standardize their use due to lack of specialized equipment and variable fabrication techniques.

To overcome these challenges, academic teams should strategize developing the organ chips in such a way that their system not only accurately represent human-like systems with robust results but is further validated to produce statistically relevant results. These systems should further be standardized in terms of their fabrication, characterization, and testing protocols, making it available to larger research groups around the world. Such systems can then be validated by multiple groups, considering the demographic characteristics and genetic variables. This will allow the translation of results from an academic setting to larger pharmaceutical companies, allowing them to run parallelized, high-throughput experiments to prove clinical relevance to in vivo models. In addition to these considerations, future advancements of OoAC devices should consider the utilization of easy-to-fabricate, mechanically tuneable, economical biomaterials with minimal to no absorption. With the advances in bench-side 3D printing devices, 3D bioprinting and 4D bioprinting, it is now possible to integrate spatial tissue configuration and biophysical characteristics using a time-dependent external stimulus. Making these advances easily accessible and versatile to use will allow the researchers and companies to microfabricate these integrated OoAC devices in significant numbers. This will allow studies to be conducted with an improved throughput with standardized operating protocols to produce comparable and clinically relevant results to the in vivo studies. Regardless, the process of bringing such a paradigm shift is quite complex and it could take several years of carefully orchestrated studies with multidisciplinary collaborations among researchers, pharma capitalists, and regulatory authorities.

## Figures and Tables

**Figure 1 ijms-24-03232-f001:**
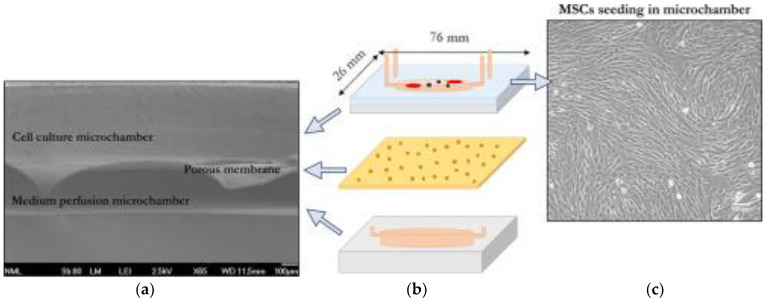
Bone marrow-on-a-chip system. (From left to right). (**a**) SEM image of cross section of the BMoC device. (**b**) 3D illustration of BMoC device with cells cultured in a porous membrane, sealed with a glass slide. (**c**) Mesenchymal stem cells cultured in microchambers of BMoC. The microchambers, coated with collagen type I, presented an organized cell expansion in stromal tissue, a well-developed 3D matrix. This figure has been adapted from a study by Tserepi et al. [15].

**Figure 2 ijms-24-03232-f002:**
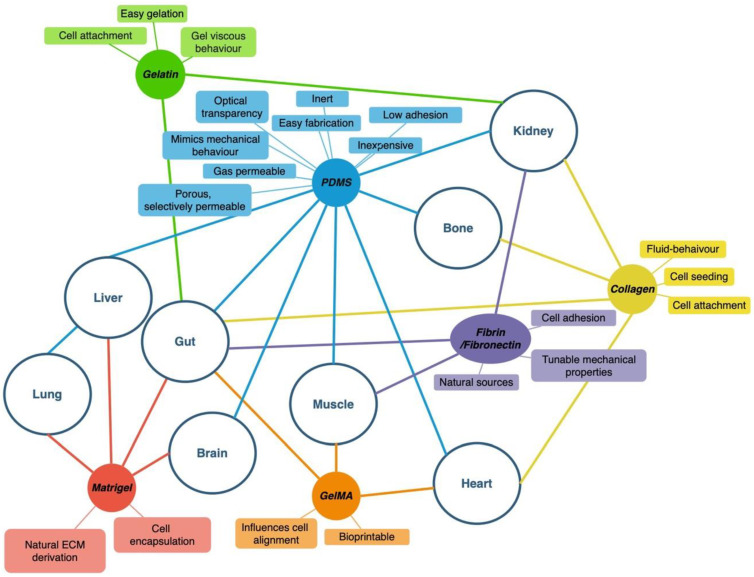
Various OoAC systems with their commonly used biomaterials (coloured circle) and associated properties (coloured square). Each coloured line connects a specific biomaterial to their associated OoAC system and properties: green for gelatin, blue for PDMS, yellow for collagen, purple for fibrin/fibronectin, orange for GelMA, and red for Matrigel.

**Table 1 ijms-24-03232-t001:** Common natural biomaterials and their biomaterial selection criteria.

Name	Biocompatibility Biodegradability	Mechanical Characteristics	Sterilization Methods	Surface Treatment	Examples of Fabricated Tissues
Collagen	Solubility changes depending on the extraction source [26] “Superior biocompatibility” [2] Enzymatic degradability [2] Cell-adhesion sites [2]	Lack of mechanical strength and structural stability when hydrated [27] Tunable [28] Young’s Modulus: 0.13–9.1 kPa [29,30] Fracture toughness (pure, in air): ~0.55 MPa m^1/2^ [31] Fracture toughness (pure, ‘in-aqua): ~0.086 ~0.55 MPa m^1/2^ (‘in-aqua’) [31]	Standard sterilization methods cause denaturation [32,33] Ethylene oxide [32,34] Gamma-radiation [32]	Collagen-coating on other biomaterials for cell adhesion [2,7,35,36]	Gut-on-a-Chip [26] Bone-on-a-Chip [37]
Gelatin	Biologically compatible and biodegradable [38] Molecular derivative of collagen [38] Cell attachment substrate for cell culture [38]	Readily forms a gel [25] Tunable degree of crosslinking [39] Young’s Modulus: 10–15 kPa [40] Varies with gelatin concentration, collagen source, and solvent composition [41]	Autoclave decreases physical properties and does not significantly inactivate endotoxins [42] Ethylene oxide [42]	Gelatin-coating on other biomaterials for cell adhesion [38]	Vascular chip [43] Muscle model [44]
Fibrin	Scaffold for cell encapsulation [2] Biocompatible, rapid biodegradability, easy fabrication [45]	Can improve mechanical properties via composition or crosslinking [45] Poor mechanical properties for skeletal tissue regeneration [45] Young’s Modulus (fibre, uncrosslinked): ~1.7 MPa [46] Fracture strain (fibre, uncrosslinked): 226% [46]	Heat and ionizing energy-based sterilization methods alter the material properties [33] Ethylene oxide effectiveness is dependent on cross-linking technique [33]	Fibrin-coating on other materials for cell adhesion [2]	Vascular tissues [47] Lung tissues [48] Skeletal muscles [49]
Hyaluronic acid A	Biocompatible, biodegradable, bioabsorbable [50] Binds to cell receptors to initiate intracellular signalling cascades [51] Modifiable to obtain stable derivatives resistant to degradation [52]	Biomechanical stability [25] Required to be crosslinked with other synthetic polymers to enhance gel-forming properties [53] Tunable viscoelastic properties [42] Mechanical characteristics depends on hydrogel formation, crosslinking, components, and constituent properties [54,55]	Autoclaving reduces physical properties [42] Ethylene oxide [42]	Can be chemically modified to increase hydrophobicity and processed into fibres, microspheres, and membranes [25]	Tissue barrier [56]
Chitosan	Biocompatible and biodegradable with antibacterial and antifungal activity [57]	Absorbs water without compromising structural stability [57] Dependent on charge and degree of crosslinking [58] Young’s Modulus: 5–2500 kPa [59] Highly dependent on weight percentage, gelation method, and degree of acetylation [26] Crosslinking increases Young’s modulus and stiffness [26]	Gamma/beta irradiation degrades chitosan [20] Limited use in dry state [60] Autoclavable [60]	Can create different scaffold types by processing (gels, nanofibers, sponges) [57]	Vascular chip [58]
Alginate	Not naturally degradable in mammals [61] Biocompatibility affected by purity [62]	Depends on the season and growth conditions of the source [63] Molecular weight can increase Young’s modulus [64] Maximum viscosity between 3 and 3.5 pH [26] Young’s modulus: 0.2–1.3 kPa but can reach up to 6kPa when weight percentage is increased [30,64]	Filtration followed by lyophilization [65] Ethanol sterilization [66] Autoclavable but may affect physical properties [67]	Lack of cell adhesion motifs, often combined with other polymers [68]	Cardiac tissue [26] Liver tissue [26]

**Table 2 ijms-24-03232-t002:** Common techniques and materials used for the fabrication of microfluidic devices.

Fabrication Techniques	Materials Used	Advantages	Limitations
Photo- lithography	- Resin [105] - Glass [106] - Elastomers(e.g., PDMS [107] and polyurethane [108]) - Thermoplastics (e.g., poly (methyl methacrylate) [109] - Cyclic olefin polymers [110]	- High precision over feature geometry [111] - Comparatively fast [107,112] - Used on different materials such as glass, and silicon [112]	- Require a cleanroom for fabrication [111] - High initial tooling and machinery costs [111] - Work only on perfectly flat substrates [113]
Soft lithography	- PDMS [114] - Liquid metal [115] - Polystyrene [116]	- Capable of mass production of sophisticated microstructures [114] - Cost-effective [117] - Relatively easy setup [118] - High throughput [118] - Cleanroom-free operation [117]	- Need aid from another lithography method to fabricate the stamp master [107]
Injection moulding	- Polystyrene [119,120] - Polymethylmethacrylate (PMMA) [121] - Cyclic olefin copolymer (COC) [121]	- High repeatability and reliability once the mould tool is made [122] - Capable of mass production at a low cost [123] - Relatively short cycle time [123] - Allows for complex geometries with tight tolerances [124] - Little plastic waste [125]	- High initial tooling and machinery costs [122] - Long initial lead times [122] - Difficult and expensive for design changes [125]
Hot Embossing process	- Polymethylmethacrylate (PMMA) [126,127] - Polycarbonate (PC) [126] - Cyclic olefin copolymer (COC) [128] - Polystyrene (PS) [129] - Polylactic acid (PLA) [130]	- Cost-effective [131] - High efficiency [131] - Capable of mass replication [132] - High structural precision [133]	- Difficult to control the optimal process parameters [134] - Require thermoplastic materials to be used [131] - Great care needed when de-embossing the products [132].
Etching Technique	- Poly (2-hydroxyethyl methacrylate) (p-HEMA) [135] - Silk fibroin [136,137] - PDMS [138]	- Capable of mass production of fine-detailed and complex microdevices [139] - Rapid Prototyping [140] - High precision fabrication [141] - Production of burr-free parts with no stress to the metal [141]	- High initial tooling and machinery costs [142] - Require high expertise to operate specialized equipment [142] - Require using some corrosive gases [143,144]
Laser cutting process	- Paper [145,146]	- Non-contact carving method [147] - Excellent accuracy and precision [148,149] - A relatively short development time [149]	- Expensive initial and maintenance costs [148] - Produce harmful pyrolysis by-products when burning plastics [147,148] - Require compressed air or water wash after finishing [148,150]
3D printing	- PDMS [151] - Glass [152,153] - Poly (ethylene glycol) (PEG) [154] - Natural biopolymers [155] (e.g., collagen [156], silk fibroin, gelatin [157], alginate, and chitosan)	- Flexible designing and modelling [158] - Excellent precision, smooth surface resolution and watertight tolerances depending on printing types [93,159] - Utilize various materials [160] - Rapid prototyping [93,160] - Minimal waste [161] - Cost-effective [162]	- Restricted building part sizes [158] - Require post-processing methods such as water jetting, sanding, chemical soaking and air drying [158,159]

## Data Availability

Not applicable.

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
