# Peer review of "Microfluidic Organ-on-A-chip: A Guide to Biomaterial Choice and Fabrication"

_ijms, 2023, doi:10.3390/ijms24043232_

Round 1

Reviewer 1 Report

this review about organ on a chip materials and fabrication is comprehensive and well written. it should provide a good reference for those unfamiliar with this field.

it could be helpful for the main shortcomings of each technique to be discussed along with the entry of the particular technique or material, but perhaps it was a style choice to save the critical review of each system's imitations until the very end in the conclusion. in any case, the critical information is at least, if not more important, than the often generalized "this material/technique has shown promise and is being studied for applications in xyz" statements that conclude many of the sections. the authors should consider balancing each section with the known disadvantages since readers don't often read conclusion sections thoroughly.

Author Response

Dear Reviewer,

We sincerely thank you for your time and valuable comments to improve our manuscript. Please find our point-by-point response to your comments below:

Comment: “It could be helpful for the main shortcomings of each technique to be discussed along with the entry of the particular technique …The authors should consider balancing each section with the known disadvantages since readers don't often read conclusion sections thoroughly.”

Response: We created a table to gather the information related to the fabrication techniques, including their advantages/limitations, and which materials go with that technique. We hope this way would be easier for the readers to understand and summarize the information. Please refer to Table 2 for more details.

Reviewer 2 Report

As a review paper, it is not too attractive since there are only one image and one table to illustrate an argument or to summarize some results and observations. Add SEM image of the microfluidic channel would be very helpful for the readers

For each section, it should be interesting to establish a comparative table or graph as a summary of the studies indicated, showing the advantages and variables used by each one for a better understanding. 

In general, it should be indicated which fabrication technique as well as the material being used for a better understanding of the results

The conclusions are too brief for this paper. A summary of the influence of each section discussed should be indicated as well as a personal opinion of the authors on the best conditions to obtain the best results.

Author Response

Dear Reviewer,

We sincerely thank you for your time and valuable comments to improve our manuscript. Please find our response to your comment below:

Comment: “As a review paper, it is not too attractive since there are only one image and one table to illustrate an argument or to summarize some results and observations. Adding an SEM image of the microfluidic channel would be very helpful for the readers."

Response: Thank you for your suggestion. We have now incorporated an SEM image of the microfluidic chamber along with a schematic illustration in figure 1.

Comment: “For each section, it should be interesting to establish a comparative table or graph as a summary of the studies indicated, showing the advantages and variables used by each one for a better understanding. In general, it should be indicated which fabrication technique as well as the material being used for a better understanding of the results.”

Response: We created a new table to gather information related to the fabrication techniques, including their advantages/limitations, and which materials go with that technique. We hope this way would be easier for the readers to understand and summarize the information. Please refer to Table 2.

Comment: "The conclusions are too brief for this paper. A summary of the influence of each section discussed should be indicated as well as an opinion of the authors on the best conditions to obtain the best results."

Response: We added additional paragraphs in our conclusions and included our opinion on the influence of the microfluidic system, its current challenges, and its outlook. Please refer to lines 557 to 590. As for the summary of sections, we have now added two informative tables (Table 1 & Table 2) to provide comments and details on the influence of each biomaterial and bio-fabrication technique for microfluidic applications.

Reviewer 3 Report

Great work! The article is very well written and provides a very detailed and comprehensive overview of the fabrication of microfluidic systems. I do have a couple of suggestions which I think will add value:

1. Include more figures especially for the section on fabrication techniques. It would be great if there is a representative figure for each of the method described. It might be even better if there are microscopic images of cells cultured in microfluidic systems that can be included to make it more appealing to readers. 

2. Please make Table 1 more comprehensive by providing what tissues are primarily fabricated using the biomaterials listed in the table. Also for the mechanical properties, can actual values be provided for parameters like elastic modulus, tensile strength, fracture toughness etc?

Use the table below as a reference for the tissue engineering applications for each hydrogel.

(Ref: Li, Xiaomeng, Qingqing Sun, Qian Li, Naoki Kawazoe, and Guoping Chen. "Functional hydrogels with tunable structures and properties for tissue engineering applications." Frontiers in chemistry 6 (2018): 499)

Author Response

Dear Reviewer,

We sincerely thank you for your time and valuable comments to improve our manuscript. Please find our response to your comment below:

Comment: “Include more figures, especially for the section on fabrication techniques. It would be great if there is a representative figure for each of the methods described. It might be even better if there are microscopic images of cells cultured in microfluidic systems that can be included to make it more appealing to readers.”

Response: Thank you for your suggestion. We created a table to gather key information related to the fabrication techniques, including their advantages/limitations, and which materials go with that technique. Also, we now show microscopic images that represent a typical microfluidic platform for cell culture application. We hope this way would be easier for the readers to understand and summarize the information. However, due to the limit on the length of a review paper, we couldn’t provide representative figures to all of the fabrication techniques. Please kindly refer to Figure 1 and Table 2 for the added information.

Comment: “Please make Table 1 more comprehensive by providing what tissues are primarily fabricated using the biomaterials listed in the table. Also, for the mechanical properties, can actual values be provided for parameters like elastic modulus, tensile strength, fracture toughness?”

Response: We provided examples of fabricated tissues by the related biomaterials. Moreover, we added values of some mechanical properties, especially young’s modulus values. Please refer to the highlighted parts in Table 1.

Round 2

Reviewer 2 Report

The paper can be accepted as it is.